# The *Sclerotinia sclerotiorum* ADP-Ribosylation Factor 6 Plays an Essential Role in Abiotic Stress Response and Fungal Virulence to Host Plants

**DOI:** 10.3390/jof10010012

**Published:** 2023-12-25

**Authors:** Kunmei Wang, Siyi Wang, Ting Wang, Qi Xia, Shitou Xia

**Affiliations:** Hunan Provincial Key Laboratory of Phytohormones and Growth Development, College of Bioscience and Biotechnology, Hunan Agricultural University, Changsha 410128, China; yuhunan@stu.hunau.edu.cn (K.W.); wangsiyi@stu.hunau.edu.cn (S.W.); tina@stu.hunau.edu.cn (T.W.);

**Keywords:** *SsArf6*, hyphal growth, appressorium formation, stress response, virulence, melanin

## Abstract

The ADP-ribosylation factor 6 (Arf6), as the only member of the Arf family III protein, has been extensively studied for its diverse biological functions in animals. Previously, the Arf6 protein in *Magnaporthe oryzae* was found to be crucial for endocytosis and polarity establishment during asexual development. However, its role remains unclear in *S. sclerotiorum*. Here, we identified and characterized *SsArf6* in *S. sclerotiorum* using a reverse genetic approach. Deletion of *SsArf6* impaired hyphal growth and development and produced more branches. Interestingly, knockout of *SsArf6* resulted in an augmented tolerance of *S. sclerotiorum* towards oxidative stress, and increased its sensitivity towards osmotic stress, indicative of the different roles of *SsArf6* in various stress responses. Simultaneously, *SsArf6* deletion led to an elevation in melanin accumulation. Moreover, the appressorium formation was severely impaired, and fungal virulence to host plants was significantly reduced. Overall, our findings demonstrate the essential role of *SsArf6* in hyphal development, stress responses, appressorium formation, and fungal virulence to host plants.

## 1. Introduction

*Sclerotinia sclerotiorum*, as a notorious soilborne plant pathogenic fungus, which has an extremely broad host range, is capable of infecting economically significant crops such as rapeseed, sunflower, and soybeans, resulting in substantial agricultural losses [1,2,3,4,5]. In China, one of the main diseases of oilseed rape is Sclerotinia stem rot (SSR) caused by *S. sclerotiorum*, leading to losses of approximate 8.4 billion RMB annually [6,7]. *S. sclerotiorum* produces sclerotia that can survive in the field for multiple years [8,9]. In addition, it employs different infection pathways, including infection of host plant tissues through the hypha by germination of sclerotia. When conditions are favorable, sclerotia can also germinate to form apothecia, releasing many ascospores to infect the hosts [1,10]. 

Previous studies indicate that the interaction mechanisms between *S. sclerotiorum* and its host are intricate. To establish successful colonization, *S. sclerotiorum* undergoes specific morphological changes in its hyphal tips, leading to the formation of appressoria [11,12,13]. Appressoria play a crucial role in adhering to the host surface and penetrating the cuticle [14]. In the past few decades, some proteins related to the development and formation of appressoria have been identified and characterized in *S. sclerotiorum*, including mitogen-activated protein kinase (MAPK) signaling pathway proteins *Ss*Ste12 [15] and *Ss*Fkh1 [16], autophagy-related proteins *Ss*ATG8, *Ss*NBR1 [17], *Ss*FoxE3 [18], and *Ss*Atg1 [19], as well as TOR signaling pathway protein *Ss*TOR [20]. Additionally, secreted proteins *Ss*Rhs1 [21], *Ss*ams2 [22], *Ss*Nsd1 [23], and *Magnaporthe* appressoria-specific (MAS) protein *Ss*cnd1 [24] have also been confirmed to participate in appressoria formation.

*S. sclerotiorum* possesses pathogenic characteristics that involve a transition from biotrophic to necrotrophic lifestyles [25]. During the early stages of infection, *S. sclerotiorum* secretes virulent factors, for instance oxalic acid [26,27], cell wall-degrading enzymes [28,29], and effectors [30,31] to impede mechanisms for recognizing and defending against pathogens [32]. Then the *S. sclerotiorum* quickly enters into the necrotrophic phase, characterized by the production of substantial amounts of reactive oxygen species (ROS) and virulent factors, which induce rapid cell death and the progression of necrotic symptoms [3,14,25,32,33]. Despite research reports identifying candidate genes [34] or significant quantitative trait loci (QTL) [35] associated with resistance to SSR in *B. napus*, there is still a lack of resistant varieties in production. As a result, chemical methods remain the primary means of controlling SSR in *B. napus* [36]. However, *S. sclerotiorum* has been reported to develop resistance to fungicides [37,38]. Furthermore, it has been observed that there are variations in the resistance of *S. sclerotiorum* to fungicides, and these variations exist both within and among populations of the fungus [39]. This has raised significant attention towards the control and prevention of *S. sclerotiorum*. 

ADP-ribosylation factor 6 (Arf6) is a member of the ADP-ribosylation factors (Arf) family, which can regulate endomembrane recycling and actin cytoskeleton remodeling at the cell surface [40,41]. In mammals, Arf6 plays a crucial role in regulating neutrophil energy metabolism [42], cancer cell invasion, metastasis, and proliferation [43], as well as membrane lipid homeostasis [44]. Additionally, a homolog of Arf6 in *Aspergillus nidulans*, named ArfB, functions in endocytosis to play important roles in polarity establishment during isotropic growth and polarity maintenance during hyphal extension [45]. The Arf6’s homolog in *M. oryzae* also has similar functions [41]. However, the biological function of Arf6 in *S. sclerotiorum* remains unclear. 

Through reverse genetic approaches, we characterized the roles of *SsArf6* in *S. sclerotiorum*, and found that *SsArf6* is highly conserved among various plant pathogenic fungi. The knockout of *SsArf6* led to hindered mycelial growth and development, characterized by increased branching and aerial hyphae formation, as well as reduced sclerotia production. Surprisingly, *SsArf6* negatively regulates melanin accumulation and hydrogen peroxide resistance. In addition, deletion of *SsArf6* significantly increased the sensitivity of *S. sclerotiorum* to osmotic stress. Most importantly, the appressorium formation and virulence to host plants exhibited severe impairments in knockout mutants. Together, our results suggest that *SsArf6* plays a significant role in the formation of appressorium, hyphal development, resistance to abiotic stress in *S. sclerotiorum*, and fungal virulence to host plants.

## 2. Materials and Methods

### 2.1. Fungal Strains and Culture Conditions

The wild-type strain *S. sclerotiorum* 1980, knockout mutant, and complemented strains were grown on potato dextrose agar plates (potato dipping powder 5 g/L, glucose 20 g/L, agar 15 g/L, and chloramphenicol 0.1 g/L, Bio-Way Technology, Shanghai, China), and cultured in a constant temperature incubator at 20 °C. The PDA plates were supplemented with a concentration of 200 μg/mL hygromycin B (Roche, Basel, Switzerland) and 100 μg/mL G418 sulfate (Geneticin, Yeasen, Shanghai, China) for long-term storage of knockout mutants and complementation strains at 4 °C.

### 2.2. Plant Materials and Growth Conditions

*Arabidopsis thaliana* (ecotype Col-0), *Brassica napus* (Zhongshuang 11), and *Nicotiana benthamiana* LAB seedlings were cultivated in a growth room at a temperature of 22 °C, with a photoperiod of 16 h of light followed by 8 h of darkness. Except for *B. napus*, which were cultivated for 5 weeks, all other plants were cultivated for 4 weeks and were selected for the virulence assays with *S. sclerotiorum.*

### 2.3. Phylogenetic Tree Construction and Sequence Analysis

Homologous sequences of *Arf6* in *Zymoseptoria tritici*, *S. sclerotiorum*, *Ustilago maydis*, *Botrytis cinerea*, *Magnaporthe oryzae*, *Fusarium oxysporum*, and *Fusarium graminearum* were obtained from the NCBI database (https://www.ncbi.nlm.nih.gov/ accessed on 16 March 2023). The *Arf6* gff3 files were obtained from the ensemblFungi database (https://fungi.ensembl.org/ accessed on 16 March 2023). The phylogenetic tree was constructed using MEGA11 programs with 1000 bootstrap replicates and the neighbor joining method [46]. The gene feature was visualized with GSDS2.0 [47], and the alignment of Arf6 proteins was performed using Clustal Omega (https://www.ebi.ac.uk/Tools/msa/clustalo/ accessed on 16 March 2023) [48] and the results were viewed with Jalview Version 2 [49].

### 2.4. Knockout and Complementation of SsArf6

The split-marker method was employed to knockout the *SsArf6* gene *(*sscle_03g022330) in *S. sclerotiorum*. As previously described [50], two rounds of PCR were used to construct the replacement fragments, SsArf6-UP-HY and SsArf6-Down-YG, which were used to replace the target gene, *SsArf6*. The PCR products were cotransformed into wild-type (WT) strain protoplasts. *SsArf6* knockout transformants were selected on PDA plates containing 200 μg/mL hygromycin B, and at least three rounds of hyphae tip transfer and protoplast purification were performed to obtain knockout mutant homozygotes.

In genetic complementation, a 1746-bp fragment consisting of upstream and full-length genomic DNA of *SsArf6* was amplified specifically through PCR. The resulting fragment was individually fused with NEO fragments, which were amplified from the *pCH-EF-1* plasmid (provided by D. Jiang from Huazhong Agricultural University). The resulting PCR products were cotransformed into protoplasts of the knockout mutant strain. Validation was achieved through PCR using six sets of primers (Appendix A), and the transformed knockout and complementation strains were selected for further experimentation.

### 2.5. DNA and RNA Manipulation

The mycelium of *S. sclerotiorum* was cultivated on PDA medium containing cellophane for 48 h, and then ground into fine powder in liquid nitrogen. Genomic DNA was extracted from both WT and mutant strains using the cetyltrimethylammonium bromide (CTAB) method [51]. The extracted WT genomic DNA was used as a template for amplifying the full-length sequence and the flanking sequence of *SsArf6*. The extracted genomic DNA from the mutants was used to confirm the knockout and complementation of *SsArf6*. To evaluate the expression of *SsArf6* in the knockout and complementation strains at the transcriptional level, the mycelium was collected after 48 h of cultivation on PDA medium containing cellophane, and total RNA was extracted using the SteadyPure Plant RNA Extraction Kit (Accurate Biology, Changsha, China). The first-strand cDNA was synthesized from the total RNA using the Evo M-MLV RT-PCR Kit (Accurate Biology, Changsha, China) as the template. Semi-quantitative RT-PCR was performed with cDNA as the template for 28 PCR cycles. All primers used in this study are listed in Appendix A.

### 2.6. Colony Morphology Observation

The WT, knockout mutant (Δ*Ssarf6*), and complemented strain (*SsArf6-C*) mycelial plugs with a diameter of 5 mm were inoculated at the center of 9 cm Petri dishes containing PDA medium. The plates were incubated at a constant temperature of 20 °C, and the mycelial growth length was measured every 24 h for a period of 48 h. The average growth rate was calculated at the end of the 48 h period. The plates were further incubated for a total of 14 days, and the number of sclerotia in each PDA plate was counted. The colony and sclerotia morphologies of the WT, Δ*ssarf6*, and *SsArf6-C* strains on PDA medium were documented using a digital camera at 24 h, 48 h, 5 days, and 14 days of cultivation, respectively. The experiment was conducted independently three times, with three technical replicates performed in each trial.

### 2.7. Stress Treatment

Mycelial plugs, measuring 5 mm in diameter, from the WT, Δ*Ssarf6*, and *SsArf6-C* strains were introduced at the center of 9 cm Petri dishes that were filled with PDA medium supplemented with varying concentrations of H_2_O_2_ (5 mM, 10 mM, and 15 mM), cell wall inhibitor Congo Red (CR), 0.02% Sodium Dodecyl Sulfate (SDS), as well as the osmotic stressors 0.5 M NaCl, KCl, and 1 M sorbitol and glucose. The plates were incubated at a constant temperature of 20 °C. After 48 h, the growth radius of mycelium under various stress conditions was measured, and the growth inhibition rate of the mycelium was calculated. In addition, the morphology of the colonies was recorded using a digital camera. The inhibition rate (%) was calculated as 100 × (the colony radius of the strain on pure PDA subtracted by the colony radius of the strain under different stressors) divided by the colony radius of the strain on pure PDA. The experiment was conducted independently three times, with three technical replicates performed in each trial.

### 2.8. Analysis of Compound Appressoria, OA, and Virulence to Host Plants

The WT, Δ*Ssarf6*, and *SsArf6-C* strain mycelial plugs with a diameter of 5 mm were inoculated onto a PDA medium containing 0.005% (*w*/*v*) Bromophenol Blue. After 48 h of being placed in a constant temperature incubator set at 20 °C the color change was recorded using a digital camera, with three repeats.

To observe appressorium formation, mycelial plugs, with a diameter of 8 mm, from the WT, Δ*Ssarf6*, and *SsArf6-C* strains were inoculated onto a slide. The slides were then transferred to square Petri dishes measuring 9 cm in diameter, which contained a folded square of moistened paper towels, measuring 25 cm in diameter, saturated with 11 mL of sterile water. The Petri dishes were then placed in a constant temperature incubator at 20 °C; after 24 h of cultivation, the morphology of the compound appressoria was observed and recorded under an optical microscope (Axio Imager 2, ZEISS, Oberkochen, Germany). After a cultivation period of 48 h, the appressorium of the WT, Δ*Ssarf6*, and *SsArf6-C* strains was recorded using a digital camera, with three repeats.

To examine the morphology of the compound appressoria formed on onion epidermal cells, mycelial plugs measuring 5 mm in diameter from the WT, Δ*Ssarf6*, and *SsArf6-C* strains were inoculated onto the onion epidermis and transferred to square Petri dishes measuring 9 cm in diameter, which contained a folded paper towel measuring 25 cm in diameter and saturated with 11 mL of sterile water, and were then cultured for 16 h at 20 °C in a temperature incubator. Afterwards, the onion epidermis was soaked in a 0.5% Trypan Blue solution for 30 min. Then, a bleaching solution was prepared in a ratio of ethanol: acetic acid: glycerol of 3:1:1, used to decolorize the samples. The morphology of the composite attached cells was then observed and recorded under an optical microscope, as previously described [52]. The experiment was conducted independently three times.

For the virulence analysis, mycelial plugs with a diameter of 2 mm and 5 mm were obtained from the wild-type (WT), Δ*Ssarf6*, and *SsArf6-C* strains. These plugs, with a diameter of 2 mm, were subsequently used to inoculate detached leaves of *A. thaliana* (5 mm in diameter on *B. napus* and *N. benthamiana*). Similarly, mycelial plugs measuring 5 mm in diameter from the corresponding strains were used to inoculate wounded detached leaves of *N. benthamiana* and were placed in square Petri dishes measuring 9 cm in diameter, which contained a folded square moistened paper towel, measuring 25 cm in diameter, saturated with 11 mL of sterile water. In a growth room at a temperature of 22 °C, with a photoperiod of 16 h of light followed by 8 h of darkness, after 36 h of infection, the infection morphology was recorded using a digital camera, and the lesion areas were analyzed using ImageJ 1.46r software to calculate the reduction in the lesion areas [53]. The reduction rate of lesion areas (%) was calculated as 100 × (the infection area of WT on detached leaves subtracted by the infection area of the Δ*Ssarf6* mutant on detached leaves) divided by the infection area of WT on detached leaves. The experiment was conducted independently three times, with three technical replicates performed in each trial.

## 3. Results

### 3.1. Identification of the Arf6 Homolog in S. sclerotiorum

Previously, when using a forward genetic approach to screen hypo-virulent mutants of *S. sclerotiorum*, we obtained one mutant with a candidate gene which was speculated to be a guanine exchange factor for Arf6 (*Ss*EFA6, sscle_12g090120), related to the pathogenic capacity of *S. sclerotiorum* (unpublished data). To determine whether Arf6 is also associated with pathogenicity, we then performed a genomic blasting and identified sscle_03g022330 as the homologous gene of *Mo*Arf6 in *S. sclerotiorum*, named *Ss*Arf6, which comprised four exons and three introns, spanning a length of 794 base pairs and encoding 186 amino acids. As shown in Figure 1A, *Ss*Arf6 homologous proteins were prevalent among plant pathogenic fungi. When conducting gene structure analysis of these homologues, we discovered that UmArf6 contained only one CDS region, while *Ss*Arf6 and *Bc*Arf6 had up to four CDSs (Figure 1B). In addition, protein sequence alignment revealed evolutionary conservation of *Ss*Arf6, which had a high similarity to Arf6 homologs in *F. oxysporum*, *F. graminearum*, *B. cinerea*, *Z. tritici*, and *U. maydis*, with 86.02%, 84.95%, 85.48%, 98.39%, 76.92%, and 76.84% amino acid sequence identity, respectively (Figure 1C).

### 3.2. Knockout of SsArf6 Leads to Aberrant Mycelium Growth, Increased Melanin Accumulation, and Decreased Sclerotium Production

In order to investigate the biological function of *SsArf6* in *S. sclerotiorum*, we generated the *SsArf6* deletion mutant, Δ*Ssarf6* (Appendix A), using a split-marker method based on homologous recombination. The results indicated that *SsArf6* was completely replaced by *hph* (Appendix A). The growth of Δ*Ssarf6* strain mycelium was hindered, with an average growth rate of 1.27 cm/24 h, significantly lower than that of the WT strain (Figure 2A,B). In addition, the Δ*Ssarf6* strains produced more branches and melanin compared to WT strains (Figure 2C,D). After incubation on PDA medium for 14 days, there were noticeable differences in colony morphology among WT and Δ*Ssarf6* strains. Specifically, the Δ*Ssarf6* strain formed many aerial hyphae (Figure 2E) but fewer sclerotia (Figure 2F,G) compared to WT. Furthermore, the phenotype of *SsArf6-C*, which was generated through in situ complementation in the Δ*Ssarf6* background using a knock-in method (Appendix A), was similar to WT (Figure 2). These findings strongly indicate that *SsArf6* plays an important role in hyphal growth, melanin accumulation, and sclerotium production.

### 3.3. SsArf6 Is Implicated in the Abiotic Stress Response in S. sclerotiorum

To explore the response of *SsArf6* to different stresses, *S. sclerotiorum* was treated with various cell wall inhibitors and osmotic stresses. When 0.5 M NaCl, 0.5 M KCl, 1 M glucose, and 1 M sorbitol were applied exogenously, the colony radii of Δ*Ssarf6* were significantly reduced, and the inhibition rate were significantly higher than that of WT and *SsArf6-C* strains (Figure 3A,B). However, in the medium supplemented with cell wall inhibitors Congo red (CR) and Sodium Dodecyl Sulfate (SDS), Δ*Ssarf6* demonstrated a stress response that was comparable to both the WT and *SsArf6-C* strains (Figure 3C,D). This implies that *SsArf6* is necessary for responding to osmotic stress in *S. sclerotiorum*, but it may not be essential for maintaining cell wall integrity.

Former studies suggested that fungal melanin is frequently located intracellularly and has antioxidant properties, as it can neutralize reactive oxygen species and other free radicals [54]. Therefore, we assessed the antioxidant capacity of the Δ*Ssarf6* mutant, and found that it exhibited significant tolerance to hydrogen peroxide compared to the WT and *SsArf6-C* strains (Figure 4A,B), indicative of the essential role of *SsArf6* in regulating the oxidative stress responses in *S. sclerotiorum*.

### 3.4. SsArf6 Is Involved in Compound Appressoria Development

To successfully colonize, *S. sclerotiorum* undergoes specific morphological changes in its hyphal tips to form appressoria to break down the host cuticle [14]. Therefore, we analyzed the formation of compound appressoria in Δ*Ssarf6* mutants and found that the production of compound appressoria by Δ*Ssarf6* mutants was significantly abnormal compared to the WT and *SsArf6-C* strains. Upon observing the formation of compound appressoria on glass slides 24 h later, we discovered that both WT and *SsArf6-C* strains formed compound appressoria on the glass surface, whereas the Δ*Ssarf6* mutant did not (Figure 5A). To confirm whether the deletion of *SsArf6* would delay the development and formation of compound appressoria, we observed the formation of compound appressoria on glass slides again after 48 h. We found that the WT and *SsArf6-C* strains were able to form a large number of compound appressoria on the glass surface, while the Δ*Ssarf6* mutant still failed to form them (Figure 5B). Additionally, similar results were obtained on onion epidermal cells (Figure 5C). We also assessed the secretion capacity of Δ*Ssarf6* mutants for oxalic acid and revealed that the secretion of oxalic acid was normal in the Δ*Ssarf6* strain, with no significant differences compared to the WT and *SsArf6-C* strains (Figure 5D). These findings indicate that *SsArf6* does not participate in the secretion process of oxalic acid in *S. sclerotium* but is crucial for the formation of compound appressoria and might be associated with the pathogenicity of *S. sclerotium*.

### 3.5. SsArf6 Is Essential for Virulence to Host Plants

To confirm whether *SsArf6* is related to the pathogenicity of *S. sclerotiorum*, we inoculated WT, Δ*Ssarf6*, and *SsArf6-C* strains onto detached leaves of *A. thaliana*, *B. napus*, and *N. Benthamian*. After 36 h of infection, the Δ*Ssarf6*, WT, and *SsArf6-C* strains caused water-soaked pale brown lesions on the leaves. However, the Δ*Ssarf6* mutants showed a significant decrease in lesion area, compared to WT and *SsArf6-C* strains, on leaf blades of different hosts (Figure 6A,B). When we further calculated the reduction rate of lesion areas infected by the Δ*Ssarf6* mutant and WT strain on detached leaves of these three host plants, the results showed that there was a difference in the reduction rate of lesion areas between *B. napus* and *A. thaliana* or *N. Benthamian*, but the difference was not significant (Figure 6C). In addition, when inoculated onto the wounded *N. Benthamiana* detached leaves, we observed a significant increase in the infection area of Δ*Ssarf6* after 36 h, compared to the intact detached leaves under the same infection conditions, but the infection area of Δ*Ssarf6* was still significantly less than that of WT and *SsArf6-C* strains (Figure 6D,E). These findings strongly indicate that *SsArf6 p*lays an essential role in the fungal virulence to hosts, except for its function in appressoria formation of *S. sclerotiorum*.

## 4. Discussion

Arf6 is a conserved protein that plays a vital role in the development of fungi; deletion of *Arf6* in *M. oryzae* and *A. nidulans* results in slower mycelium growth and an increased number of mycelial branches [41,45]. In this study, we identified a homologous protein *Ss*Arf6 in *S. sclerotiorum*, which shares a high degree of similarity with Arf6 proteins in other plant pathogenic fungi. Knockout of *SsArf6* led to impaired hyphal development, increased branching and melanin accumulation, and excessive growth of aerial hyphae, as well as negatively impacted sclerotia yield in *S. sclerotiorum*.

Fungal melanin, as a potent antioxidant, protects cells by scavenging hydrogen peroxide, hydroxyl radicals, and superoxide anions [55]. To investigate whether or not the *SsArf6* mutant is tolerant to oxidative stress due to its melanin accumulation, we simulated oxidative stress by adding different concentrations of hydrogen peroxide exogenously to evaluate the antioxidant capacity of the *SsArf6* deletion mutant. As expected, Δ*Ssarf6* mutants indeed exhibited significant tolerance to oxidative stress caused by hydrogen peroxide, compared to WT and *SsArf6-C*. Additionally, we observed that Δ*Ssarf6* mutants were more sensitive to hyperosmotic stress but unaffected by cell wall inhibitory agents, indicating that *SsArf6* plays an important but different role in responding to different abiotic stress.

Appressoria and oxalic acid are essential for the interaction between *S. sclerotiorum* and its host. The formation of appressoria helps to break down the physical barriers of the host, such as the cell wall and cuticle [56], and oxalic acid is a key virulence factor in the invasion process, as its secretion enhances the activity of hydrolytic enzymes [26], induces programmed cell death in plants [57], and inhibits host defenses [27]. The absence of *SsArf6* resulted in abnormal appressoria development while it did not affect oxalic acid secretion in *S. sclerotiorum*, suggesting that *SsArf6* is involved in the interaction between *S. sclerotiorum* and its host in a manner other than the oxalic acid pathway.

In *M. oryzae*, Arf6 was reported to be nonessential for its pathogenicity [41]. When fungal virulence of the *SsArf6* deletion mutant was assessed, we observed significantly decreased virulence compared to the WT and *SsArf6-C* strains. Furthermore, when infection assays were performed on wounded leaves of *N. benthamiana*, the area of infection by the Δ*Ssarf6* mutant significantly increased compared to unwounded leaves 36 h post inoculation, although it still remained significantly lower than that of the WT and *SsArf6-C* strains under the same infection conditions. This evidence indicated an essential role of *SsArf6* in fungal virulence to hosts, except for its function in appressoria formation of *S. sclerotiorum*. Further characterization is needed regarding the specific mechanisms by which *SsArf6* regulates the formation of appressoria and pathogenicity.

Taken together, *SsArf6* is involved in mycelial growth, appressorium development, and stress response in *S. sclerotiorum* and contributes to the infection process and fungal virulence to host plants. Our study provides evidence for an improved understanding of the role of Arf6 in the interaction between *S. sclerotiorum* and its hosts.

## Figures and Tables

**Figure 1 jof-10-00012-f001:**
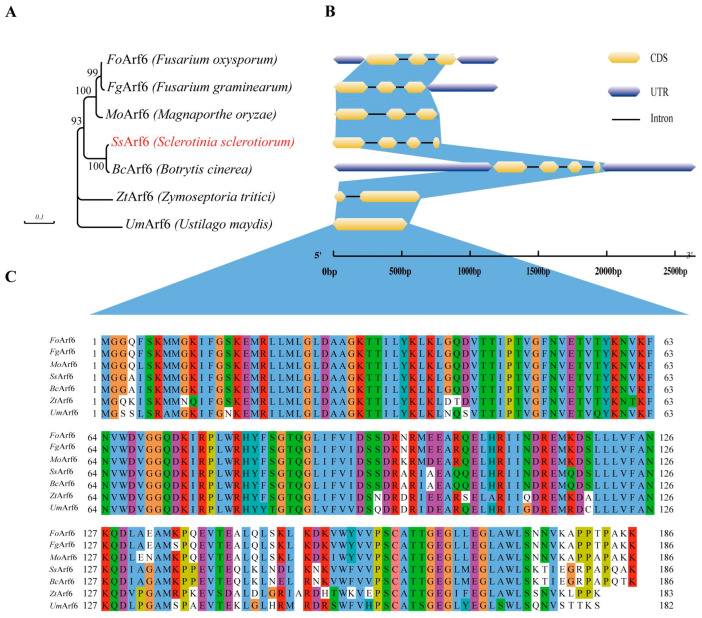
Phylogenetic and sequence analysis of *SsArf6*. (**A**) Phylogenetic analysis of Arf6 between different species. (**B**) The *Arf6* gene features in *F. oxysporum*, *F. graminearum*, *M. oryzae*, *S. sclerotiorum*, *B. cinerea*, *Z. tritici*, and *U. maydis* illustrated schematically by GSDS2.0. (**C**) Multiple sequence alignment of *Fo*Arf6, *Fg*Arf6, *Po*Arf6, *Ss*Arf6, *Bc*Arf6, *Zt*Arf6, and *Um*Arf6. The alignment results were visualized using Jalview Version 2 clustal Colour Scheme. Blue indicates hydrophobic amino acid, Red: Positive charged amino acid, Magenta: Negative charged amino acid, Green: Polar amino acid, Pink: Cysteines, Orange: Glycines, Yellow: Prolines, Cyan: Aromatic amino acid, White: Unconserved amino acid.

**Figure 2 jof-10-00012-f002:**
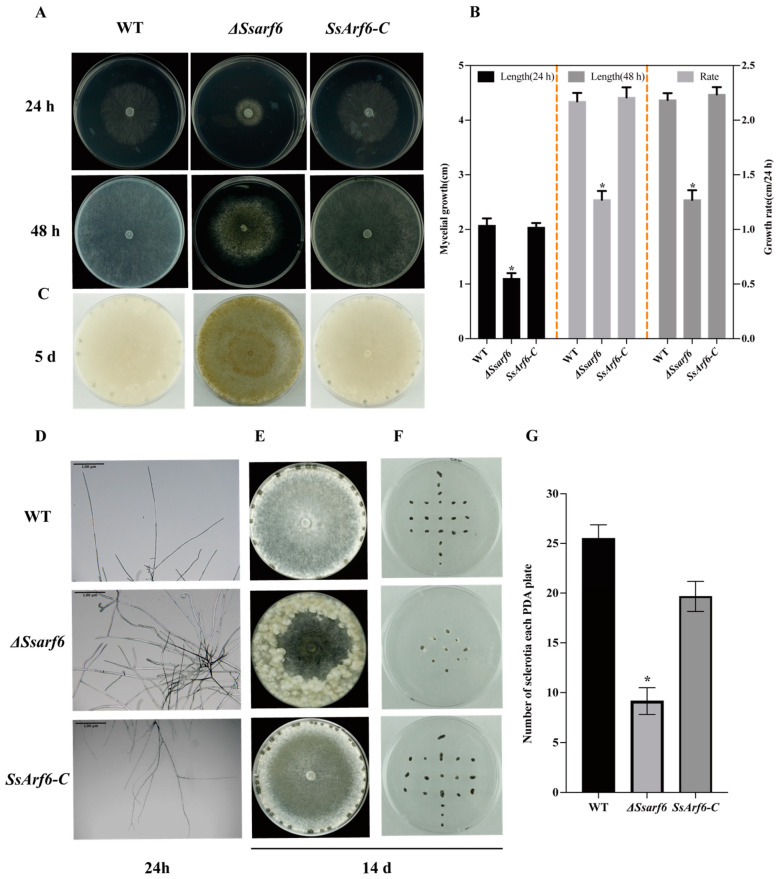
*SsArf6* contributes to mycelial development, melanin accumulation, and sclerotia production in *S. sclerotiorum.* (**A**) The colony morphology of WT, Δ*Ssarf6*, and *SsArf6-C* strains grown on PDA culture medium for 24 h and 48 h. (**B**) The average mycelial length of WT, Δ*Ssarf6*, and *SsArf6-C* strains, measured at 24 and 48 h after inoculation, and average growth rate. (**C**) Melanin accumulation of WT, Δ*Ssarf6*, and *SsArf6-C* strains grown on PDA culture medium for 5 d. (**D**) Branching patterns of mycelia of WT, Δ*Ssarf6*, and *SsArf6-C* strains. (**E**) The colony morphology of WT, Δ*Ssarf6*, and *SsArf6-C* strains grown on PDA medium for 14 days. (**F**,**G**) The number of sclerotia per plate. WT refers to the wild-type strain; Δ*Ssarf6*, the knockout strain; and *SsArf6-C*, the complemented strain. The experiment was repeated three times with similar results. Error bars represent the standard deviation (SD). The statistical significance between WT and knockout mutant or complemented strains was analyzed using the Student’s *t*-test (* *p* < 0.05).

**Figure 3 jof-10-00012-f003:**
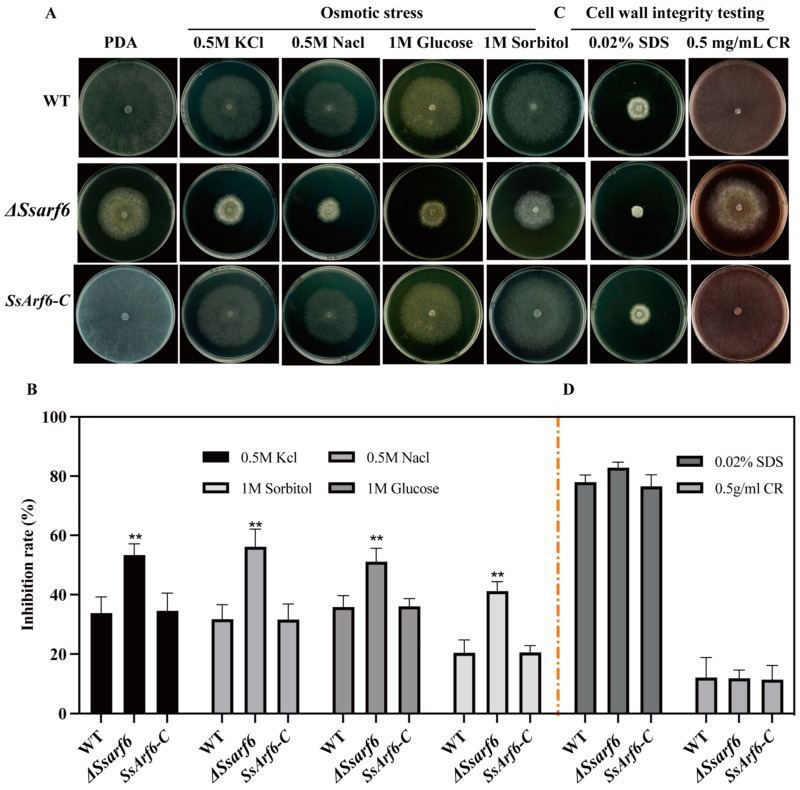
*SsArf6* contributes to the responses to osmotic stress in *S. sclerotiorum*. (**A**,**B**) The colony morphology and inhibition rate of WT, Δ*Ssarf6*, and *SsArf6-C* strains grown on PDA medium containing different osmotic stressors, 0.5 M NaCl, 0.5 M KCl, 1 M glucose, and 1 M sorbitol, for 48 h. (**C**,**D**) The colony morphology and inhibition rate of WT, Δ*Ssarf6*, and *SsArf6-C* strains grown on PDA medium containing different cell wall inhibitors, 0.5 mg/mL Congo Red (CR) and 0.02% Sodium Dodecyl Sulfate (SDS), for 48 h. Δ*Ssarf6* represents the knockout strain and *SsArf6-C*, the complemented strain. The experiment was conducted three times with similar results. Error bars represent the standard deviation (SD). The statistical significance between the WT and knockout mutant or complemented strains was analyzed using the Student’s *t*-test (** *p* < 0.01).

**Figure 4 jof-10-00012-f004:**
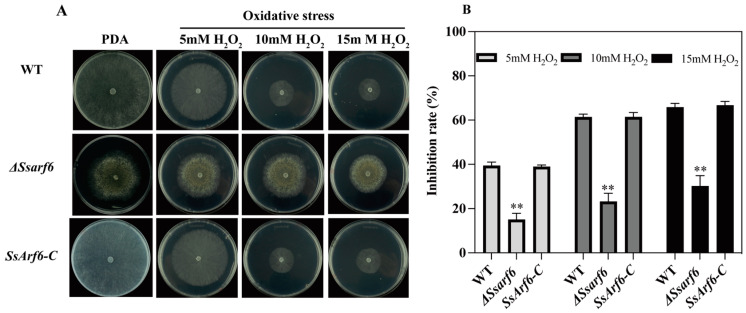
*SsArf6* negatively regulates the resistance of *S. sclerotiorum* to hydrogen peroxide. (**A**,**B**) The mycelium morphology and inhibition rate of WT, Δ*Ssarf6*, and *SsArf6-C* strains grown on PDA medium containing 5 mM, 10 mM, and 15 mM H_2_O_2_ for 48 h. The experiment was conducted three times with similar results. Error bars represent the standard deviation (SD). The statistical significance between the WT and knockout mutant or complemented strains was analyzed using the Student’s *t*-test (** *p* < 0.01).

**Figure 5 jof-10-00012-f005:**
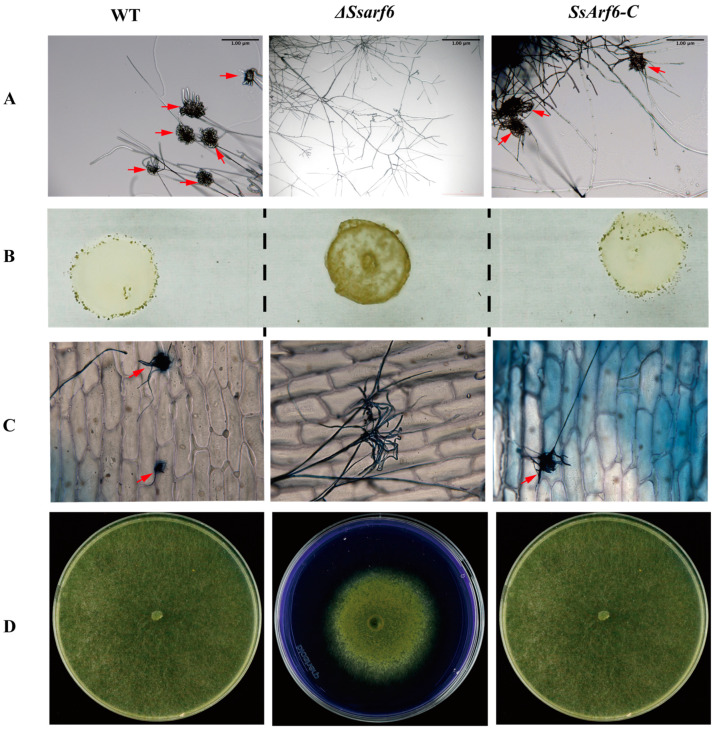
*SsArf6* is involved in compound appressoria formation. (**A**) Appressorium formed, after 24 h on glass slides, by WT, Δ*Ssarf6*, and *SsArf6-C* strains observed under an optical microscope. Bar = 1 µm. (**B**) The appressorium morphology, after 48 h on glass slides, for WT, Δ*Ssarf6*, and *SsArf6-C* strains. (**C**) Invasion assay of WT, Δ*Ssarf6*, and *SsArf6-C* on onion epidermis. Invasion mycelial were stained with Trypan Blue. (**D**) Analysis of the oxalic acid secretion ability of WT, Δ*Ssarf6*, and *SsArf6-C* strains, with colony morphology after 48 h cultivation on PDA medium containing Bromophenol Blue. When the pH of Bromophenol Blue is greater than or equal to 3.0 and less than 4.6, it appears yellow, and when the pH is greater than or equal to 4.6, it appears blue. The experiment was repeated three times with similar results. Red arrows point to appressoria. WT represents the wild-type strain; Δ*Ssarf6*, the knockout strain; and *SsArf6-C*, the complemented strain.

**Figure 6 jof-10-00012-f006:**
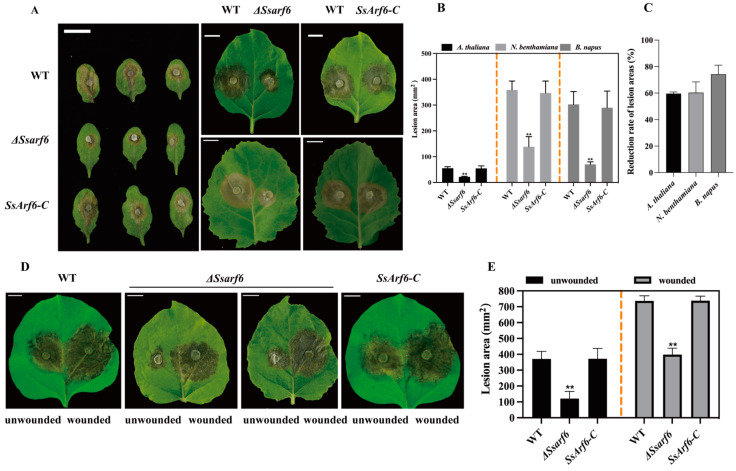
*SsArf6* contributes to the virulence of *S. sclerotiorum.* (**A**) Inoculated lesions of WT, Δ*Ssarf6*, and *Ssarf6-C* of detached leaves of *A. thaliana*, *B. napus*, and *N. benthamiana*. (**B**) Lesion areas of WT, Δ*Ssarf6*, and *SsArf6-C* on leaves of *A. thaliana*, *B. napus*, and *N. benthamiana*. (**C**) Reduction rate of lesion areas between Δ*Ssarf6* mutant and WT strain on detached leaves of *A. thaliana*, *B. napus*, and *N. benthamiana*. (**D**) Inoculated lesions of Δ*Ssarf6* on detached unwounded/wounded leaves of *N. benthamiana*. (**E**) Lesion areas of Δ*Ssarf6* on wounded/unwounded leaves of *N. benthamiana*. Data were recorded at 36 h post inoculation. Bar = 10 mm. ImageJ 1.46r was used to analyze the lesion area. The experiment was repeated three times with similar results. Error bars represent SD. The statistical significance between WT and knockout mutant or complemented strains was analyzed using the Student’s *t*-test (** *p* < 0.01). WT represents the wild-type strain; Δ*Ssarf6*, the knockout strain; and *SsArf6-C*, the complemented strain.

## Data Availability

The GenBank accession numbers (species names) for organisms used in this study are as follows: *Zymoseptoria tritici* (*Zt*Arf6, XP_003857013.1), *Sclerotinia sclerotiorum* (*Ss*Arf6, APA07463.1), *Ustilago maydis (Um*Arf6, XP_011391884.1), *Botrytis cinerea (Bc*Arf6, XP_001547581.1), *Magnaporthe oryzae* (*Mo*Arf6, XP_003715902.1), *Fusarium oxysporum* (*Fo*Arf6, XP_018253040.1), and *Fusarium graminearum* (*Fg*Arf6, XP_011321112.1) were obtained from the NCBI database (https://www.ncbi.nlm.nih.gov/ accessed on 16 March 2023). The Arf6 gff3 files were obtained from the ensemblFungi database (https://fungi.ensembl.org/ accessed on 16 March 2023).

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
