# Peer review of "The Sclerotinia sclerotiorum ADP-Ribosylation Factor 6 Plays an Essential Role in Abiotic Stress Response and Fungal Virulence to Host Plants"

_jof, 2023, doi:10.3390/jof10010012_

Round 1

Reviewer 1 Report

Comments and Suggestions for Authors

Jof-2750422

Authors

1.- The research is very well developed and structured

2.- on line 87 specify the number of plants

3.- In the methodology, specify if an experimental design was used.

Author Response

We genuinely thank the expert reviewers for your constructive and helpful suggestions and comments, which help us to improve the quality of our manuscript. The manuscript has been revised substantially as suggested. Our responses in green font can be found beneath each original reviewer’s comment. A version of the revised manuscript with track changes is included for review.

We believe this revised version is much improved, and hope the manuscript is now suitable for publication in JOF.

1. on line 87 specify the number of plants

Reply: We had cultivated a significant number of plants for various experiments. In this study, a total of 30 Arabidopsis thaliana (ecotype Col-0) plants, 16 Brassica napus (Zhongshuang 11) plants, and 24 Nicotiana benthamiana plants were used.

2. In the methodology, specify if an experimental design was used.

Reply: We sincerely appreciate your valuable suggestion, and made revisions accordingly, please see the details in the manuscript:

lines 136-146 in manuscript 2.6., lines 148-169 in manuscript 2.7., and lines 171-205 in manuscript 2.8.

Reviewer 2 Report

Comments and Suggestions for Authors

The paper is interesting and seems that SsArf6plays an important role in hyphal growth, melanin synthesis and sclerotium production. However, as the description of the tests has several problems, the reality of the findings is not easy to find out. For this reason, I suggest first a major revision, and depending on that, we can decide about the publication. .

Line 50. Does it mean that S. sclerotiorum behaves in the first several days as a hemibiotrophic organism?  Is it proven, or this paper delivers proofs for that? When it is known, please cite the article where this was published. If this is a new result, please treat it accordingly.

Line 55. We worked earlier with S. sclerotiorum in sunflower. We have found resistance differences. It is very probable that immunity will not be found, but smaller and useful differences in resistance can be found. Possibly, for this the inoculation methods should be made more precise to verify smaller differences in resistance.  It would be also interesting when not only 1 isolate per variant would have been checked, but 2-3 isolates with the same genetic background, and their means would represent the genetic differences between plants much better than a single isolate whose aggressiveness can vary significantly. 

Line 85. Please give the variety name or the genotype code for identification. Genotype differences between cultivars can be very significant. To repeat the test with these genotypes needs an exact characterization of the genotypes.

Line 142. Fort the inoculation tests an experimental design should be added, inoculation method should be described, number of repetitions of the test should also be reported. How many leaves were tested for a variant within a replicate. For example,  in a test 20 leaves were treated. Anthis was repeated three times as reported in the discussion.  Inoculum production should also be described or saying that the medium was the same described in 2.1. on PDA without hygromycin adding. Can you say something about the fungal material concentration in the inoculation tests described in 2.8.? Significant concentration differences can lead to aggressiveness differences between inocula. In such tests this is a very sensitive area. In Fusarium a diluting and a concentrating action can significantly increase or decrease the aggressiveness of the inoculum and so the experimental results. In this case we should be sure that the concentration of the inoculum was the same for the wild type and knocked out version securing that the infection severity differences do not come from concentration differences of the inoculum.

This is not the first case that I see a beautifully planned genetical and DNA work, but the quality of the plant side and inoculation  is not considered as important as should be. This is what I see here. The possibility that the work was made correctly, but not described as it should have been. When this is the case, the improvement and presenting the missing information is not a problem. In the other case this may question the reality of the presented results. My opinion is that any molecular and other genetic work is reasonable when the classic experimental conditions are correctly followed and demonstrated. 

Reviewer 3 Report

Comments and Suggestions for Authors

Dear Colleagues. There are several questions regarding the methodology.
Sincerely

Round 2

Reviewer 2 Report

Comments and Suggestions for Authors

Dear Authors, 

I think it was worth to rewrite the Materials and Methods. So thi is OK now. There are several points to improve, I suggest minor corrections. The new comments are printed in bold. 

Response to the Comments of Reviewer 2

We genuinely thank the expert reviewers for your constructive and helpful suggestions and comments, which help us to improve the quality of our manuscript. The manuscript has been revised substantially as suggested. Our responses in green font can be found beneath each original reviewer’s comment. A version of the revised manuscript with track changes is included for review.

We believe this revised version is much improved, and hope the manuscript is now suitable for publication in JOF.

1. Line 50. Does it mean that S. sclerotiorum behaves in the first several days as a hemibiotrophic organism? Is it proven, or this paper delivers proofs for that? When it is known, please cite the article where this was published. If this is a new result, please treat it accordingly.

Reply: Thanks for your excellent question. Indeed, there is a nutritive growth phase during the interaction between S. sclerotiorum and host plants, and the duration of this phase may vary depending on the host. Mehdi Kabbage's study revealed a close association between the extracellular space of S. sclerotiorum mycelium and onion epidermal cells (Fig 1A and B), and it was observed that the mycelium thickens considerably upon entry into the extracellular matrix without immediately killing the plant cells. Furthermore, infection experiments conducted on tobacco revealed that, compared to the oxalate-deficient mutants A2 and B. cinerea (C), the wild-type strain of S. sclerotiorum 1980 still allowed host cell survival within the infection area (Figure 2). Based on these findings, Mehdi Kabbage and colleagues proposed A new model depicting the lifestyle transition of S. sclerotiorum ( Fig 3).The content of this work has been cited within the manuscript, as evidenced by the citation of reference 25.

Fig 1 Biotrophic growth of S. sclerotiorum on onion epidermal layer. S. sclerotiorum (strain 1980) was inoculated onto onion epidermal cells and placed on a microscope slide. 24 h post inoculation; tissue was stained with Trypan blue. (A)to reveal fungal hyphae and determine viability of onion cells. Thick invasive hypha (compared to aerial hyphae) is shown growing within living tissue. Sucrose induced plasmolysis was used to assess cell viability (B).

Comment: My question was what is the case with the hemi-biotrophic nature of the parthogenic nature of S. sclerotiorum. In the answer you gave a good comment, but there bis no word about it in the improved text. When this fits, you should include a sentense with the Kabbage study  and his 1st figure that this is the case. When you are not sure, you can say that we have data on this (with citation), but we need more data to prove it. Mí point is some response should be given to this remark also in the tex, not only in the answer to the reviewer.

2. Line 55. We worked earlier with S. sclerotiorum in sunflower. We have found resistance differences. It is very probable that immunity will not be found, but smaller and useful differences in resistance can be found. Possibly, for this the inoculation methods should be made more precise to verify smaller differences in resistance. It would be also interesting when not only 1 isolate per variant would have been checked, but 2-3 isolates with the same genetic background, and their means would represent the genetic differences between plants much better than a single isolate whose aggressiveness can vary significantly.

Reply: Thanks for your question and suggestions. In this study, stringent conditions were applied to ensure the accuracy of the infection experiments, including precise control of the duration of plant cultivation and the process of infection. For the virulence analysis, mycelial plugs with a diameter of 2 or 5 mm were obtained from the wild-type (WT), ΔSsarf6, and SsArf6-C strains. These plugs with a diameter of 2 mm were subsequently used to inoculate detached

Fig 3 A new model depicting the lifestyle transition of S. sclerotiorum. S. sclerotiorum grows within the apoplastic space without crossing plant cell wall during the early stages of infection, while host cells remain alive. We propose that during this time, the fungus secretes oxalic acid and other pathogenicity factors that modulate host cell defense responses. The fungus then quickly switches to necrotrophic growth leaving a trail of dead cells, while biotrophy is maintained at the leading edge of fungal colonization.

Fig 2 Growth habits of S. sclerotiorum and B. cinerea in tobacco leaf tissue. Agar plugs containing actively growing cultures of wild type S. sclerotiorum (strain 1980) (A), S. sclerotiorum oxalate deficient A2 mutant (B), and B. cinerea (C) were inoculated onto tobacco leaves. 24 h post-inoculation, leaf tissue was stained with Trypan blue to assess tissue viability.

leaves of A. thaliana (5 mm in diameter on B. napus and N. benthamiana) Similarly, mycelial plugs measuring 5 mm in diameter from the corresponding strains were used to inoculate wounded detached leaves of N. benthamiana, placed on square petri dishes measuring 9 cm in diameter, which contained a folded 25 cm in diameter square moistened paper towels saturated with 11 mL of sterile water. In a growth room at a temperature of 22

℃, with a photoperiod of 16 hours of light followed by 8 hours of darkness, after 36 hours of infection, the infection morphology was recorded using a digital camera, and the lesion areas are analyzed using Image J software. The experiment was conducted independently three times, with three technical replicates performed in each trial. Thus, we believe the differences of lesion areas are true under the stringent conditions, and reflect the differences in resistance.

Comment: My question was, whether do you have resistance differences between genotypes or not. You stated that there are no resistance to S. sclerotiorum. In this respect we need several citations who states that, because this is a very severe statement. The other question is what do you consider on resistance?  When the immunity is resistance, you are probably right. resistance. When you have two genotypes having across locations, years or isolates significant difference, that is difference in resistance response, this can be determined by QTLs as happens with a number of polygenically inherited resistance traits. For this we need a clear-cut answer,

3. Line 85. Please give the variety name or the genotype code for identification. Genotype differences between cultivars can be very significant. To repeat the test with these genotypes needs an exact characterization of the genotypes.

Reply: Thanks you for your suggestion. The names of corresponding species have been added to line 85 of the manuscript.

Comment. For Nicotiana benthamiana has possible no varieties (I do not know), but you might have an identification number from a gene bank when somebody would like to repeat the test. should know where and how to get.

Line 136. The petri dish is Petri dish, as Dr. Petri made first this lab dish and therefore it was termed by his name.

4. Line 142. Fort the inoculation tests an experimental design should be added, inoculation method should be described, number of repetitions of the test should also be reported. How many leaves were tested for a variant within a replicate. For example, in a test 20 leaves were treated. Anthis was repeated three times as reported in the discussion. Inoculum production should also be described or saying that the medium was the same described in 2.1. on PDA without hygromycin adding. Can you say something about the fungal material concentration in the inoculation tests described in 2.8.? Significant concentration differences can lead to aggressiveness differences between inocula. In such tests this is a very sensitive area. In Fusarium a diluting and a concentrating action can significantly increase or decrease the aggressiveness of the inoculum and so the experimental results. In this case we should be sure that the concentration of the inoculum was the same for the wild type and knocked out version securing that the infection severity differences do not come from concentration differences of the inoculum.Line 142. Fort the inoculation tests an experimental design should be added, inoculation method should be described, number of repetitions of the test should also be reported. How many leaves were tested for a variant within a replicate. For example, in a test 20 leaves were treated. Anthis was repeated three times as reported in the discussion. Inoculum production should also be described or saying that the medium was the same described in 2.1. on PDA without hygromycin adding.

Chapter 2.6. is OK.

Chapter 2.7. is OK.

Reply: Thanks for your suggestion. Based on your comments, modifications have been made to lines 77~83 and lines194-205 of the manuscript, providing

detailed descriptions of the inoculum dosage, cultivation conditions, composition and concentration of PDA culture medium, detection methods, and number of experimental replicates for each strain. Please see the details in the manuscript.

2.8. Line 161. Please explain OA.

5. Can you say something about the fungal material concentration in the inoculation tests described in 2.8.? Significant concentration differences can lead to aggressiveness differences between inocula. In such tests this is a very sensitive area. In Fusarium a diluting and a concentrating action can significantly increase or decrease the aggressiveness of the inoculum and so the experimental results. In this case we should be sure that the concentration of the inoculum was the same for the wild type and knocked out version securing that the infection severity differences do not come from concentration differences of the inoculum.

Reply: Yes. The PDA formulation employed in this research consisted of Potato dipping powder 5 g/L, Glucose 20 g/L, Agar 15 g/L, and chloramphenicol 0.1 g/L, which was sourced from Bio-Way Technology in Shanghai, China. in each trial

Line 317. It would be better to set arrows for appressoriums( or appressoria) in A version. When I see the dark mycelium (?) mass in the figures A, the appressoria cannot be identified well. Thes are very similar to the invasion assays in C pictures.

Line 344. It is clear that the mutant has a much lower virulence. But there is another  thing that is important. The three plants differ significantly in resistance. A thaliana is much more resistant than the other two plants are. You said in the beginning that no resistance exists Ss. This result does not prove it.  For this reason it would be important to analyze results also in this aspect. It would increase the value of the paper.

As a Summary, the improvement of the methodical part made reliable the results achieved. I support now the publication, but the paper should be improved and the resistance relations discussed in more detail.
